# Permissive Modulation of Sphingosine-1-Phosphate-Enhanced Intracellular Calcium on BK_Ca_ Channel of Chromaffin Cells

**DOI:** 10.3390/ijms22042175

**Published:** 2021-02-22

**Authors:** Adonis Z. Wu, Tzu-Lun Ohn, Ren-Jay Shei, Huei-Fang Wu, Yong-Cyuan Chen, Hsiang-Chun Lee, Dao-Fu Dai, Sheng-Nan Wu

**Affiliations:** 1Krannert Institute of Cardiology and Division of Cardiology, Department of Medicine, Indiana University School of Medicine, Indianapolis, IN 46202, USA; 2Department of Physiology, National Cheng-Kung University Medical College, Tainan 701401, Taiwan; 3Institute of Zoology, National Taiwan University, Taipei 106319, Taiwan; tzu-lun.wang@helmholtz-muenchen.de (T.-L.O.); joyoyo@ibms.sinica.edu.tw (Y.-C.C.); 4Institute of Tissue Engineering & Regenerative Medicine, Helmholtz Zentrum München, 85764 Neuherberg, Germany; 5Division of Pulmonary, Allergy, and Critical Care Medicine, Department of Medicine and Gregory Fleming James Cystic Fibrosis Research Center, University of Alabama at Birmingham, Birmingham, AL 35294, USA; reshei@alumni.iu.edu; 6Neuroscience Program of Academia Sinica, Academia Sinica, Taipei 115201, Taiwan; hfwu@gate.sinica.edu.tw; 7Institute of Molecular Biology, Academia Sinica, Taipei 115201, Taiwan; 8Institute of Biomedical Sciences, Academia Sinica, Taipei 115201, Taiwan; 9Division of Cardiology, Department of Internal Medicine, Kaohsiung Medical University Hospital, Kaohsiung Medical University, Kaohsiung 807378, Taiwan; hclee@kmu.edu.tw; 10School of Medicine, College of Medicine, Kaohsiung Medical University, Kaohsiung 807378, Taiwan; 11Graduate Institute of Clinical Medicine, Kaohsiung Medical University, Kaohsiung 807378, Taiwan; 12Department of Pathology, Carver College of Medicine, University of Iowa, Iowa City, IA 52242, USA

**Keywords:** sphingosine-1-phosphate, BK_Ca_ channel, intracellular Ca^2+^, chromaffin cell

## Abstract

Sphingosine-1-phosphate (S1P), is a signaling sphingolipid which acts as a bioactive lipid mediator. We assessed whether S1P had multiplex effects in regulating the large-conductance Ca^2+^-activated K^+^ channel (BK_Ca_) in catecholamine-secreting chromaffin cells. Using multiple patch-clamp modes, Ca^2+^ imaging, and computational modeling, we evaluated the effects of S1P on the Ca^2+^-activated K^+^ currents (*I*_K(Ca)_) in bovine adrenal chromaffin cells and in a pheochromocytoma cell line (PC12). In outside-out patches, the open probability of BK_Ca_ channel was reduced with a mean-closed time increment, but without a conductance change in response to a low-concentration S1P (1 µM). The intracellular Ca^2+^ concentration (Ca_i_) was elevated in response to a high-dose (10 µM) but not low-dose of S1P. The single-channel activity of BK_Ca_ was also enhanced by S1P (10 µM) in the cell-attached recording of chromaffin cells. In the whole-cell voltage-clamp, a low-dose S1P (1 µM) suppressed *I*_K(Ca)_, whereas a high-dose S1P (10 µM) produced a biphasic response in the amplitude of *I*_K(Ca)_, i.e., an initial decrease followed by a sustained increase. The S1P-induced *I*_K(Ca)_ enhancement was abolished by BAPTA. Current-clamp studies showed that S1P (1 µM) increased the action potential (AP) firing. Simulation data revealed that the decreased BK_Ca_ conductance leads to increased AP firings in a modeling chromaffin cell. Over a similar dosage range, S1P (1 µM) inhibited *I*_K(Ca)_ and the permissive role of S1P on the BK_Ca_ activity was also effectively observed in the PC12 cell system. The S1P-mediated *I*_K(Ca)_ stimulation may result from the elevated Ca_i_, whereas the inhibition of BK_Ca_ activity by S1P appears to be direct. By the differentiated tailoring BK_Ca_ channel function, S1P can modulate stimulus-secretion coupling in chromaffin cells.

## 1. Introduction

Sphingosine-1-phosphate (S1P), a bioactive sphingolipid, is a blood constituent which elicits numerous biological responses associated with immune cell trafficking [1]. S1P is released by activated platelets [2] and acts as an autocrine or paracrine factor to control many essential cellular functions [3], ranging from a rapid endothelial nitric oxide secretory response to long-term cell growth or death [4,5]. As an extracellular ligand, S1P has been shown to regulate the activity of ionic channels in various cell types. These actions include blocking the Kv1.3 channel in T lymphocytes [6], activating the inward-rectifying K^+^ current in atrial cardiomyocytes [7], eliciting transient receptor potential channels in vascular smooth muscle cells [8], as well as activating the large-conductance Ca^2+^-activated K^+^ channel (BK_Ca_) in endothelial cells [9]. BK_Ca_ is sensitive to both increased intracellular Ca^2+^ (Ca_i_) and/or depolarized membrane potentials (V_m_), providing an interplay between the metabolic and electrical state of the cellular activities [10,11]. In conjunction with postganglionic neurons, adrenal chromaffin cells release catecholamine to coordinate the sympathetic response to both environmental and internal stressors. In chromaffin cells, the BK_Ca_ activity is critical for controlling exocytosis by altering V_m_ depolarization and/or Ca_i_ mobilization [12,13]. However, whether BK_Ca_ is modulated by S1P in these catecholamine-secreting cells remains elusive and the underlying mechanisms are still unknown. In the current study, we performed electrophysiological measurements combined with mathematical modeling to characterize the roles of S1P in BK_Ca_ modulation in chromaffin cells. Our data demonstrate that S1P exerts a biphasic effect on BK_Ca_ activities through different mechanistic regulations.

## 2. Results

### 2.1. S1P Decrease the Single-Channel Currents of BK_Ca_ in Cell-Free Mode

We used outside-out patch recording, the cell-free mode, to evaluate the direct effect of S1P on BK_Ca_ activities of chromaffin cells bathed in a high-K^+^ bathing solution containing 1.8 mM CaCl_2_. S1P (1 µM) significantly reduced the open probability of BK_Ca_ channel without changing the single-channel conductance (Figure 1). In contrast, S1P (1 µM) did not have any effects on the BK_Ca_ activity in inside-out patches. Figure 1D shows the relationship between the S1P concentration and the percentage inhibition of BK_Ca_ open probability. The application of S1P (0.3−30 µM) was found to decrease the probability of channel openings in a concentration-dependent manner. Fitting the dose-response curve with the Hill equation yielded a half-inhibitory concentration of 1.1 µM. In addition, a slope coefficient of 1.2. S1P at a concentration of 30 µM nearly abolished the BK_Ca_ activity. The effect of S1P (1 µM) on the kinetics of BK_Ca_ gating was further analyzed. In outside-out patches of control, closed time histograms at the level of +60 mV were fitted by a two-exponential curve (Figure 2A). The time constant for the fast and slow components of the closed time histogram were 116 ± 11 and 45 ± 9 ms, respectively (*n* = 6). When applied to the bath, S1P (1 µM) significantly increased the mean closed times to 167 ± 18 and 63 ± 11 ms (*n* = 6, *p* < 0.05). Figure 2B shows the simulated single-channel data generated using transitional rates of BK_Ca_ obtained from the experiments. Based on the modeling scheme, the equilibrium dissociation constants in the control and S1P were estimated to be 3.6 ± 0.3 and 3.9 ± 0.3 (*n* = 5), whereas the equilibrium gating constants in the control and S1P were 0.20 ± 0.07 and 0.033 ± 0.009 (*n* = 5), respectively. The results suggested that the outer membrane exposure to S1P caused a significant decrease in the gating constant (by about six-folds), but no change in the dissociation constant.

### 2.2. High-Dose S1P Elevate Intracellular Ca^2+^ (Ca_i_) Thereby Enhancing BK_Ca_ Activities in On-Cell Patches

Next, we measured Ca_i_ transients in fura-2-loaded chromaffin cells. The cells were bathed in a normal Tyrode solution with 1.8 mM CaCl_2_. A representative example of the fura-2 ratio, R (340/380 nm), during the exposure to S1P is shown in Figure 3. S1P (1 µM) did not alter the Ca_i_ level, however, a high-dose S1P (10 µM) significantly induced the Ca_i_ elevation. An application of high K^+^ (45 mM) further elevated the Ca_i_ intensity. The results indicate that a low-dose of S1P had little or no effect but a high-dose of S1P (10 µM) enhanced Ca_i_. In the cell-attached configuration (on-cell mode), a high-dose S1P (10 µM) significantly increased the opening of BK_Ca_ channel (Figure 3C). The open probability of BK_Ca_ channel at +60 mV in the absence of S1P was 0.029 ± 0.002 (*n* = 8). One minute after the high-dose S1P, the open probability was significantly increased to 0.084 ± 0.003 (*p* < 0.05, *n* = 12). These channel activities were suppressed by paxilline (Pax, 1 µM), a selective BK_Ca_ channel blocker. The single-channel current amplitude and BK_Ca_ conductance were unaltered in all the groups, suggesting that BK_Ca_ can be induced by Ca_i_ at a high depolarization potential (i.e., +60 mV).

### 2.3. Biphasic Effects of S1P on I_K(Ca)_ of Chromaffin Cells

We evaluated the effects of S1P on *I_K(Ca)_* of chromaffin cells by whole-cell voltage-clamp recording. The cells were bathed in a normal Tyrode solution with 1.8 mM CaCl_2_, and the pipette solution containing 0.1 µM Ca^2+^. Within one minute of exposing the cells to S1P (1 µM), the *I*_K(Ca)_ was suppressed throughout the entire range of depolarization steps. Representative *I*_K(Ca)_ traces and averaged current-voltage (*I-V*) relations for *I*_K(Ca)_ in the control, during exposure to S1P (1 µM) and washout of S1P are illustrated in Figure 4A, respectively. S1P (1 µM) significantly suppressed *I*_K(Ca)_, however, this inhibitory effect was readily reversed after removal of S1P. Then, we examined the effect of S1P at a high concentration (10 µM) on *I*_K(Ca)_. When the cells were exposed to a high-dose S1P (10 µM), a biphasic response in the amplitude of *I*_K(Ca)_ was observed, i.e., an initial decrease followed by a persistent elevation (Figure 4B). In other words, there was a dual effect of high-dose S1P on *I*_K(Ca)_ that was elicited by a series of voltage pulses ranging from –30 to +50 mV in 20-mV increments. When the patch pipette was filled with 10 mM BAPTA, a Ca^2+^ chelator, S1P (10 µM) did not induce an activated *I*_K(Ca)_. The activation of *I*_K(Ca)_ by S1P is Ca_i_-dependent which needs to reach the threshold Ca_i_ level to be activated.

### 2.4. Low-Dose S1P Increase Chromaffin Cell Excitability

The effect of S1P on AP firing was investigated by the current-clamp configuration, the cells were bathed in a normal Tyrode solution with 1.8 mM CaCl_2_. The typical effect of S1P (1 µM) on APs is illustrated in Figure 5A. When cells were exposed to S1P, the firing of APs in response to the current stimuli with a duration of 1.6 s was increased. For example, a low-dose S1P significantly increased the firing frequency to 3.1 ± 0.2 Hz (*n* = 5) from a control of 1.9 ± 0.2 Hz (*n* = 6). This increase in the firing of AP may result from the *I*_K(Ca)_ inhibition in chromaffin cells. The AP firing could hardly be generated by 10 µM S1P. Next, we assessed how the decreased conductance of *I*_K(Ca)_ used to mimic the action of S1P can result in APs changes in modeled chromaffin cells. Figure 5B illustrates the time course of repetitive APs in response to the current injection which was simulated from a simulated cell model. Notably, when the maximal conductance of *I*_K(Ca)_ was arbitrarily decreased (from 0.64 to 0.32 nS), the firing rate was increased. This result reinforced our observations showing that S1P (1 µM) increased the frequency by two-fold. Concomitant with changes in repetitive firing, AP repolarization was slowed down after a reduction of *I*_K(Ca)_ conductance which was used to mimic S1P effects. 

### 2.5. S1P Modulate I_K(Ca)_ and BK_Ca_ in PC12

The effects of S1P on *I*_K(Ca)_ and BK_Ca_ of pheochromocytoma PC12 cells were further evaluated. The cells were bathed in a normal Tyrode solution with 1.8 mM CaCl_2_, and the pipette solution containing 0.1 µM Ca^2+^. In whole-cell recording, when the cells were exposed to S1P (1 µM), the *I*_K(Ca)_ was significantly reduced (Figure 6). Similar to that in chromaffin cells, the BK_Ca_ activity was observed in the on-cell patches of PC12 bathed in a high-K^+^ bathing solution with 1.8 mM CaCl_2_. In the cell-attached configuration, a high-dose S1P (10 µM) increased the open probability of BK_Ca_ channel. This stimulatory effect was not observed in a low-dose S1P (1 µM). The BK_Ca_ activities were increased in the presence of squamocin, a Ca^2+^ ionophore [14], and 10 µM S1P. When Pax (1 µM), a BK_Ca_ channel blocker, was applied to the bath, the S1P-stimulated BK_Ca_ opening was significantly reversed (Figure 7). Consistent with the recordings in chromaffin cells, S1P (10 µM) can stimulate the BK_Ca_ activity expressed in PC12, although 1 µM S1P suppresses *I*_K(Ca)_.

## 3. Discussion

The present study demonstrated novel findings that in chromaffin cells, S1P produced biphasic modulations of both inhibitory and stimulatory effects on the BK_Ca_ channel depending on different dose levels. The inhibitory effects of S1P (1 µM) on *I*_K(Ca)_ likely result from BK_Ca_ suppression due to the modified lipid component. The activation of BK_Ca_ is closely associated with the level of Ca_i_ at a high-dose S1P (10 µM). In whole-cell recording, S1P also had a bimodal effect on *I*_K(Ca)_. Our simulation data predicted that the decreased maximal conductance of the BK_Ca_ channel could replicate the experimental data showing that S1P-induced AP firing to stimulate chromaffin cells.

### 3.1. Endogenous S1P Dosage in Physiology

S1P is abundant in the blood and it can regulate the specific tissues of the whole body. Platelets, which lack the S1P digest enzyme are therefore, rich in S1P and can release S1P upon prothrombotic stimuli, such as thrombin and collagen [15]. The serum S1P concentrations range from 100 nM to 4 μM, and the variation is derived from the different detection methods and by species [16,17,18]. Mouse serum S1P, for example, was measured as 0.84 μM or >4 μM S1P in two different studies, respectively [17,19]. The human serum S1P was reported as 0.3 or 1 μM [16,18]. Moreover, the bioactive concentration of S1P can be changed by its binding lipoproteins, i.e., high- or low-density lipoprotein (HDL or LDL), respectively. The LDL-bound S1P is biologically inactive [20], whereas the HDL-bound S1P is biologically active [21]. The concentration of S1P in blood and plasma constituents are significantly higher than the estimated kDa of S1P to its receptors, which is in the range of 8 to 60 nM [22]. Here, we used the concentrations of 1 and 10 µM to mimic the bioactive status of S1P.

### 3.2. S1P Modulate BK_Ca_ Channel Kinetics in Chromaffin Cells

BK_Ca_ in chromaffin cells yielded a single-channel conductance of ~200 pS [13,23]. Under symmetrical K^+^ conditions, the BK_Ca_ activity can be observed in our study similar to previous reports [13,23]. In the outside-out patches, S1P, when applied from the extracellular side on this cell-free mode, was found to be an effective blocker of BK_Ca_. The binding site for S1P on BK_Ca_ may be located in the extracellular leaflet. S1P probably associated with its unusual solubility for a lipid due to the polar head-group for the effects of polyamines on BK_Ca_ [24]. We demonstrated that S1P can directly decrease the BK_Ca_ activity in a dose-dependent fashion with an *IC*_50_ value of 1.1 µM. This dosage is very close to normal S1P levels in the serum, which range from 0.4~0.7 µM depending on the species and detect methods differences [4,16,17,18,19]. The physiological dose of S1P had little effect on the mean open time of BK_Ca_, but produced a lengthening in the mean close time. The inhibitory effects of S1P lie primarily on the BK_Ca_ kinetics, since there is a difference in the gating constant between, before, and after the S1P application. Our findings indicate that in chromaffin cells, S1P can modify the BK_Ca_ gating kinetics leading to a decrease in *I*_K(Ca)_ for upregulating the cell excitability. Therefore, the BK_Ca_ channel can well be an important substrate for S1P in regulating these catecholamine-secreting cells.

### 3.3. S1P-Induced Ca_i_ Elevation Triggers BK_Ca_ Activation

In chromaffin cells, extracellular Ca^2+^ entrance can regulate the BK_Ca_ activity [25]. The reasons that S1P stimulates the BK_Ca_ activity possibly related to the Ca_i_ elevation via either the influx of extracellular Ca^2+^ into the cytosol through voltage-gated calcium channel (Ca_v_) currents (*I*_Ca_), or the release of Ca^2+^ from intracellular stores [26,27]. S1P at a concentration of 1 µM could not induce *I*_Ca_ in chromaffin cells [28]. The S1P-mediated stimulation of BK_Ca_ may not result from the activation of Ca_v_. However, whether S1P can activate the transient receptor potential channel remains to be further explored in the neuroendocrine cells, as previously reported [8]. Our data demonstrate that S1P increases the open probability of BK_Ca_ in the cell-attached (on-cell) configuration, whereas it produced a reduction in the BK_Ca_ activity in the outside-out (cell-free) patches. At the physiological dose level, S1P may affect the cell excitability without any change of Ca_i_. Therefore, a high-dose S1P-triggered BK_Ca_ activation can be an inhibition mechanism of Ca_i_ elevation for feedback controlling secretion in chromaffin cells.

### 3.4. Biphasic Effects of S1P on I_K(Ca)_ and the Mechanism of S1P-Mediated Stimulus-Secretion

S1P is metabolized from sphingosine by sphingosine kinases. Sphingosine was shown to have higher potency in triggering endocytosis compared with S1P [29] and the increase of neurotransmitter release has been reported in bovine chromaffin cells [29,30]. S1P regulates exocytosis through distinct mechanisms. S1P can be an extracellular agonist [4,31] and the intracellular S1P contributed a significant extent to the catecholamine release [32]. The extracellular S1P can modulate the rate of exocytosis, while the intracellular S1P may control the fusion pore expansion during exocytosis [32]. In the present study, S1P is complex with biphasic effects of suppressing *I*_K(Ca)_ at a physiological level, but increase *I*_K(Ca)_ amplitude at a high-dosage. The stimulatory effect of S1P on *I*_K(Ca)_ was related to the Ca_i_ concentration, whereas inhibitory effects were directly mediated on BK_Ca_, probably extracellularly, although the stimulatory S1P regulation results mainly from the Ca_i_ elevation. Based on the experimental observations of S1P-suppressed BK_Ca_ increase AP firing, our simulation data mimicked the excitatory of S1P on V_m_ modulation in a modeled chromaffin cell. V_m_ depolarization was reported to increase the intracellular S1P production in PC12 cells [33]. Therefore, S1P can act as both an extracellular mediator and intracellular second messenger on the stimulus-secretion coupling by tailoring BK_Ca_. When formed intracellularly in response to extracellular agonists, S1P would be released and modify cell-surface ionic channels [4,5,31,34]. The formation of intracellular S1P and the blockade of BK_Ca_ by extracellular S1P may synergistically act on catecholamine secretion of chromaffin cells.

### 3.5. Possible Mechanisms of S1P-Mediated Regulations on BK_Ca_ and Study Limitations

Our observations imply that S1P exerts a bimodal effect on *I*_K(Ca)_ through different mechanistic regulations of Ca_i_ in chromaffin cells. At the tested doses of S1P, BK_Ca_ could be modulated by different mechanisms. S1P binds to the S1P receptor (S1PR), the specific cell surface G protein-coupled receptors (GPCRs). The possible regulatory mechanisms of S1P on BK_Ca_ may encompass both S1PR-dependent [35] and GPCR-independent [9] modulations. A high-dose S1P may not selectively affect BK_Ca_ by the S1PR-dependent regulation, but could possibly regulate other K^+^ channels, such as the delayed-rectifier K^+^ current [34], or trigger the Ca_i_ elevation as we demonstrated. GTP and G proteins are necessary for the S1P function through GPCR. S1P can activate BK_Ca_ in the absence of GTP and in the presence of the G protein inhibitors [9], which suggested that S1P regulates BK_Ca_ independently of GPCR. Moreover, the fatty acid-induced modulation of BK_Ca_ channel was reported to be lipid-type dependent [36,37,38,39]. Based on these key observations, S1P at a low concentration may work through an S1PR-dependent pathway to regulate BK_Ca_ directly, whereas a high S1P may trigger multiple mechanisms, such as GPCR-independent [9] or fatty acid-induced BK_Ca_ functional modifications [37]. Nonetheless, the exact mechanism through which S1PR of S1P-mediated BK_Ca_ regulation in chromaffin cell deserves further investigation. Five subtypes of S1PRs in the cell surface have been identified, i.e., S1P1 to S1P5 [40]. S1PRs, initially called the endothelial differentiation gene receptors [41], belonged to the members of GPCRs, and it had been reported that S1P1 to S1P3 are expressed in PC12 and/or chromaffin cells [32,42,43,44]. A limitation here is that we did not directly measure the specificity of S1PRs responsible for the S1P-mediated BK_Ca_ regulation. This is beyond the scope of this study, however, our findings still provide novel insights into the involvement of S1P in the permissive modulation of Ca_i_ on the BK_Ca_ channel in the stimulus-secretion coupling of chromaffin cells.

In conclusion, the present study shows that the S1P-mediated permissive *I*_K(Ca)_ stimulation is indirect and results from an elevated Ca_i_, whereas S1P inhibits the BK_Ca_ channel activity directly. S1P can modulate the adrenal chromaffin cell excitability by the differentiated tailoring BK_Ca_ channel function.

## 4. Materials and Methods

### 4.1. Cell Preparations

Bovine adrenal glands were obtained from a local slaughterhouse and chromaffin cells were prepared by digestion with collagenase (0.5 mg/mL) followed by the density gradient centrifugation, as previously described [28]. These cells were kindly provided by Dr Chien-Yuan Pan (National Taiwan University, Taipei, Taiwan). The chromaffin cells were maintained in Dulbecco’s modified Eagle’s medium (DMEM) supplemented with 10% fetal bovine serum (FBS) in a 5% CO_2_ at 37 °C. Electrophysiological experiments were carried out between 5 and 10 days after the cells isolated. Pheochromocytoma PC12 cells were obtained from the Bioresources Collection and Research Center (BCRC-60048; Hsinchu, Taiwan) and maintained in the RPMI-1640 medium supplemented with 10% heat-inactivated horse serum and 5% FBS. All the protocols were approved by the National Cheng-Kung University Investigation Committee.

### 4.2. Drugs and Solutions

D-erythro-sphingosine-1-phosphate (S1P), ionophore, and collagenase were obtained from Sigma-Aldrich (St. Louis, MO, USA). Paxilline was from Biomol Laboratories, Inc. (Plymouth Meeting, PA, USA). BAPTA (bis-(*o*-aminophenoxy) ethane-*N,N,N′,N′*-tetraacetic acid tetrakis) and fura-2/AM were obtained from Molecular Probes. Unless otherwise stated, culture media (e.g., DMEM), FBS, horse serum, and trypsin/EDTA were acquired from Thermo Fisher (Waltham, MA, USA), while other chemicals, such as EGTA and HEPES, were of analytical grade. S1P was dissolved in a chloroform:methanol/1:19 solution to a concentration of 1 mM. Then, it was air-dried and dissolved in ethanol to make a stock of 1 mM and stored at −20 °C.

For electrophysiological recordings, the composition of all the extracellular or intracellular solutions (i.e., normal Tyrode solution) used in this work was elaborated in Table 1. In the experiments of the recording *I*_K(Ca)_ or membrane potential, we used normal Tyrode in the bath and K^+^-aspartate solution in the pipette. The pipette solution contained low EGTA (0.1 mM). For the single-channel recordings, the high K^+^ solutions were used in both external and internal milieus. The free Ca^2+^ concentration of 0.1 µM was calculated assuming a dissociation constant for EGTA and Ca^2+^ in the pipette solutions.

### 4.3. Intracellular Ca^2+^ (Ca_i_) Measurements

Cells were loaded with 5 µM fura-2/AM for 30 min at 25 °C in a normal Tyrode solution. The coverslips on which the cells were grown were mounted in a 1-mL chamber and placed on an inverted fluorescence microscope (DM-IRE2, Leica, Wetzlar, Germany). Ca_i_ were monitored with digital imaging using a C-IMAGING system equipped with a Polychrome V high-speed monochromator (Compix Inc, Sewickley, PA, USA). Fura-2/AM was excited sequentially by 340 and 380 nm light delivered from a xenon lamp via a ×20, 1.3 NA UV fluorescence objective (Leica). Fluorescent images were collected at a 510 nm scale every 1.0 s by a Peltier-cooled CCD camera. The intensity of fluorescence ratio, 340/380 nm, from individual cells was measured by the SimplePCI software (Compix). 

### 4.4. Electrophysiological Measurements

Immediately before each experiment, cells were transferred to a chamber positioned on the stage of an inverted microscope (Leica). The glass pipettes were fabricated from Kimax-51 capillaries (Kimble Glass, Vineland, NJ, USA) using a PP-830 puller (Narishige, Japan) and the tips were fire-polished with an MF-83 microforge (Narishige). When filled with a pipette solution, their resistance ranged between 3 and 5 MΩ. Ionic currents were recorded in different patch-clamp configurations (i.e., cell-attached, outside-out, and whole-cell variants), with the use of an RK-400 amplifier (Bio-Logic, Claix, France) controlled by the ClampEx program (Molecular Devices, Sunnyvale, CA, USA). To measure *I*_K(Ca)_, cells were bathed in a normal Tyrode solution with 1.8 mM CaCl_2_ at room temperature. Each cell was held at 0 mV to inactivate other non-specific voltage-gated K^+^ current. The method elicits a family of large, noisy, and outward currents with rectification. When external Ca^2+^ was removed, the current amplitude was reduced and the residue current was specifically identified as *I*_K(Ca)_.

### 4.5. Single-Channel Analyses

Single-channel currents of BK_Ca_ channel were analyzed in the pCLAMP 10.4 software (molecular devices). Multi-Gaussian adjustments of the amplitude distributions among channels were used to determine single-channel currents. The functional independence between the channels was verified by comparing the observed stationary probabilities. The probability of channel openings was evaluated using an iterative process to minimize the χ^2^ calculated with many independent observations. Open or closed lifetime distributions were fitted with a logarithmically scaled bin width. To estimate all the transition rates between states, single-channel data recorded from chromaffin cells were idealized and used to determine single-channel kinetic parameters by means of a maximum likelihood algorithm in the QUB software [45]. The highest log likelihood for the observed BK_Ca_ channel activity was obtained with the gating scheme: (1)$$C_{1}\overset{k_{-2}}{\underset{k_{2}}{⇄}}C_{2}\overset{k_{-1}}{\underset{k_{1}}{⇄}}O$$
where O is the open state, and C_1_ and C_2_ represent the closed states. On the basis of this linear scheme, single-channel data were modeled, and transition rates were obtained. The gating scheme consists of one open and two closed states. Transition rates between states were derived from maximum likelihood estimations [45]. The kinetic analysis was based on the assumption that channel states in the patch are mutually independent, and the BK_Ca_ channel can be described by a finite-state Markovian model.

To calculate the percentage decrease in the BK_Ca_ activity resulting from the presence of S1P, the potential was held at +60 mV. When outside-out patches were formed, S1P at different concentrations (0.1−30 µM) was applied to the bath. The concentration-dependent relationship of S1P on the inhibition of BK_Ca_ activity in adrenal chromaffin cells was fitted to the Hill equation using a nonlinear regression analysis. That is,
(2)$$\%\;decrease=\frac{E_{max}\cdot {\left[C\right]}^{n}}{IC_{50}{}^{n}+{\left[C\right]}^{n}}$$
where [*C*] indicates the S1P concentration; *IC*_50_ and *n* are the concentrations for a 50% inhibition and the Hill coefficient, respectively; and *E_max_* is the S1P-induced maximal inhibition of BK_Ca_ channel.

### 4.6. Statistical Analysis

Averaged results are presented as the mean ± SEM. Paired *t*-tests were used to compare two different treatments in the same cell. One-way repeated measures ANOVA with post-hoc tests were used to compare the means among three or more different treatments. A two-sided *p*-value of < 0.05 was considered as statistically significant.

### 4.7. Computer Simulations

Simulated APs in bovine chromaffin cells were run using the AP firings originally derived from gonadotropin-releasing hormone-secreting neurons [46]. The model consists of a Na^+^ current, L-type and T-type Ca^2+^ currents, a delayed rectifier K^+^ current, and an M-like current. In addition, *erg* (*ether-à-go-go*-related-gene) K^+^ and Ca^2+^-activated K^+^ currents were incorporated to this model, given that they are consistently present in chromaffin cells [47,48]. Simulations were carried out using the stochastic algorithm as implemented in the program *xpp* with the aid of XPPAUT. Source codes were downloaded from http://senselab.med.yale.edu/senselab/modeldb (22 February 2021). Expressions and parameters for ion currents, equilibrium functions, and parameter values are modified to mimic APs in adrenal chromaffin cells observed in this study. The conductance values and reversal potentials, together with other parameters, used to solve the set of all differential equations are listed in Table 2.

## Figures and Tables

**Figure 1 ijms-22-02175-f001:**
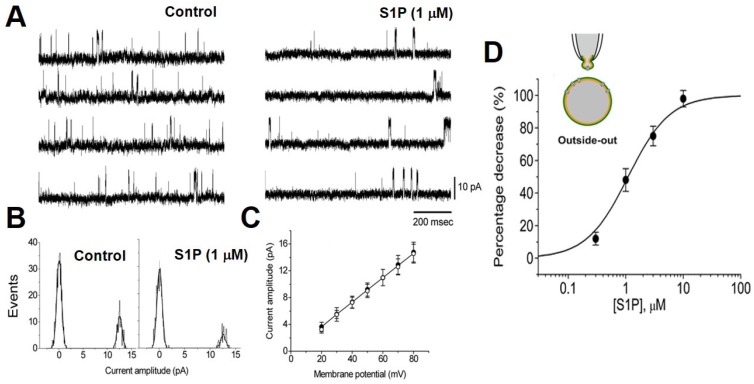
Effects of sphingosine-1-phosphate (S1P) on BK_Ca_ activities in outside-out patches. (**A**) Original traces in the control (left) and after the application of S1P (1 µM, right) in the bath. The pipette was filled with a high K^+^ solution with 0.1 µM Ca^2+^, the cells were bathed in a high-K^+^ bathing solution with 1.8 mM CaCl_2_, holding at +60 mV. (**B**) Amplitude histograms of BK_Ca_ obtained in the absence (left) and presence (right) of S1P. Data were fitted by Gaussian distributions with the maximum likelihood method. (**C**) Averaged *I-V* relations of BK_Ca_ obtained in the absence (⬤) and presence (◯) of S1P. Each point represents the mean ± SEM (*n* = 6−8). The single-channel conductance between the control and S1P is identical. (**D**) The dose-response relationship for S1P-induced inhibition of BK_Ca_. The *IC*_50_ value, maximally inhibitory percentage, and Hill coefficient were 1.1 µM, 100%, and 1.2, respectively. Each point represents the mean ± SEM (*n* = 4−9, in five cultures).

**Figure 2 ijms-22-02175-f002:**
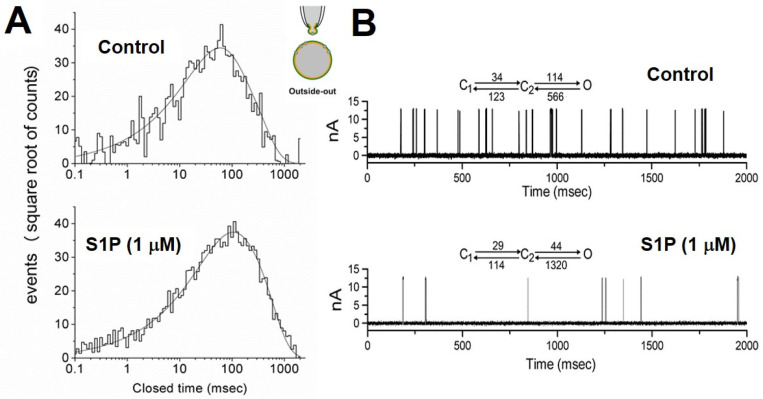
Effects of S1P on BK_Ca_ kinetics in outside-out patches (as inset). (**A**) The effects of S1P on mean closed times of BK_Ca_ at +60 mV. The pipette solution containing 0.1 µM Ca^2+^, the cells were bathed in a high-K^+^ bathing solution with 1.8 mM CaCl_2_. The closed time histograms in the control (upper) and after the application of S1P (1 µM, lower) are illustrated. The abscissa and ordinate show the logarithm of apparent open or closed times (ms) and the square root of the number of counts, respectively. (**B**) Simulated single-channel currents in the control (upper) and S1P (lower). Simulation models used to analyze the observed data measured at +60 mV are shown in the upper part of each simulated current. Each horizontal arrow pointing to the left represents the binding of S1P or closing of a channel, and each arrow to the right represents the dissociation of S1P or opening of a channel. The units are µM/ms or /ms. O: Open state; C_1_ and C_2_, two closed states.

**Figure 3 ijms-22-02175-f003:**
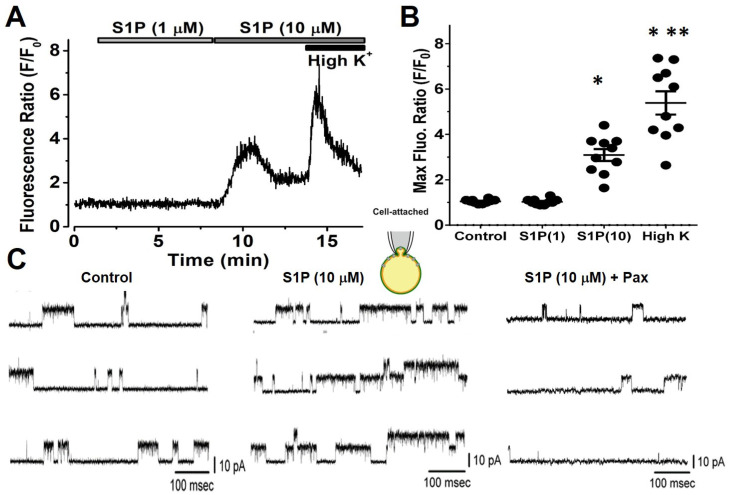
Effects of S1P (10 µM) on Ca_i_ and BK_Ca_ activation of chromaffin cells. (**A**) Representative Ca_i_ trace of the fluorescence ratio (340/380 nm) from an individual cell loaded with fura-2/AM. The cells were bathed in a normal Tyrode solution with 1.8 mM CaCl_2_. For Ca_i_, there was no change in the S1P (1 µM) application but a high-dose S1P (10 µM) elevated the Ca_i_ transients and was further increased by high K^+^ (45 mM) subsequently. (**B**) Summary of S1P effects on Ca_i_. Asterisks indicate significant differences (*p* < 0.05, * control and ** 10 µM S1P, *n* = 10, in four cultures). (**C**) S1P-enhanced BK_Ca_ on a cell-attached patch (as inset) measured at +60 mV. The cells were bathed in a high-K^+^ bathing solution with 1.8 mM CaCl_2_. BK_Ca_ events were obtained in the control (left), S1P (10 µM, middle), and after paxilline (Pax, 1 µM, right) combined with S1P which were significantly suppressed.

**Figure 4 ijms-22-02175-f004:**
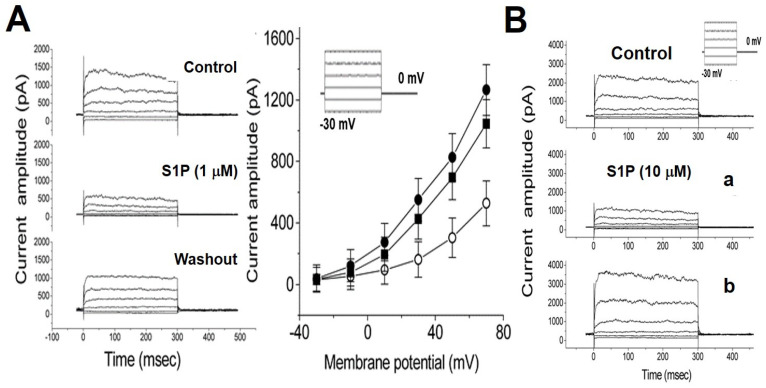
Effects of S1P on *I*_K(Ca)_ in chromaffin cells. (**A**) Superimposed current traces (left panel) obtained in the control, exposure to S1P (1 µM), and washout. The cells were bathed in a normal Tyrode solution with 1.8 mM CaCl_2_, and the pipette solution containing 0.1 µM Ca^2+^. Averaged *I-V* relationships of *I*_K(Ca)_ measured at the end of each voltage pulse (right panel) in the control (⬤), exposure to 1 µM S1P (◯), and washout (⏹). Each point represents the mean ± SEM (*n* = 6−10, in four cultures). (**B**) Biphasic effects of high-dose S1P (10 µM) on *I*_K(Ca)_. Superimposed current traces obtained when the recording pipette was filled with ethylene glycol tetraacetic acid (EGTA, 0.1 mM). The uppermost part is the control. Time courses after the S1P (10 µM) application are at 1 (a) and 3 min (b). The pulse protocol is shown as an inset part of each related graph.

**Figure 5 ijms-22-02175-f005:**
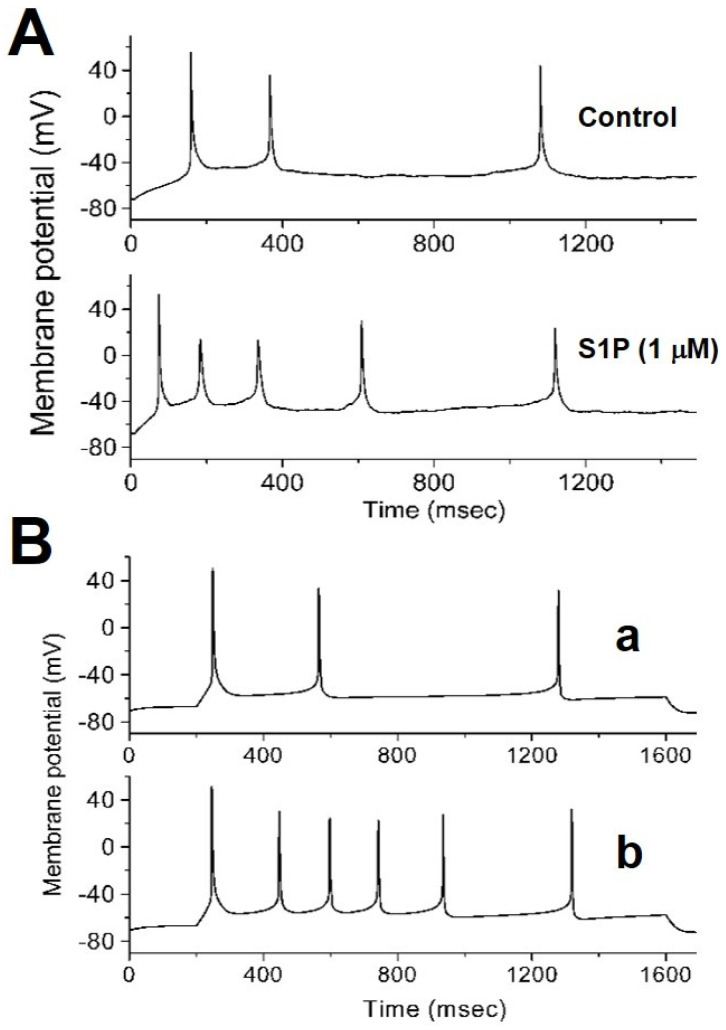
Effects of S1P on action potentials (APs) and simulated AP of a modeling chromaffin cell. (**A**) Current-clamp conditions were performed with the current injection of 2 nA applied to the cell with a duration of 1.6 s. The cells were bathed in a normal Tyrode solution with 1.8 mM CaCl_2_. Potential traces were recorded in the control and exposure to S1P (1 µM). (**B**) The simulated APs were generated from a cell model which mimics the electrical behavior of chromaffin cell. A current injection at the 4.6 nA with 1.4 s duration was applied. S1P (1 µM) was mimicked *I*_K(Ca)_ by the maximal conductance (g_K(Ca)_) from 0.64 (a) to 0.32 nS (b).

**Figure 6 ijms-22-02175-f006:**
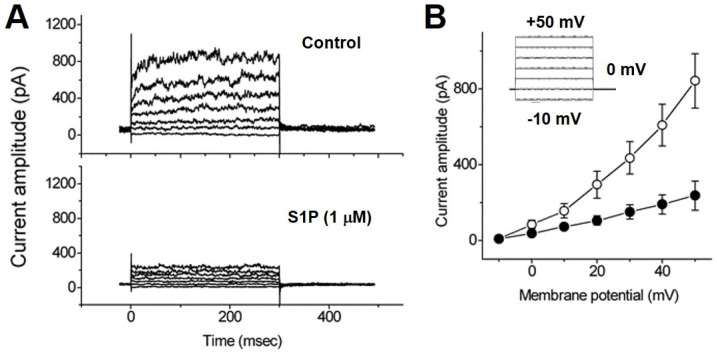
Inhibitory effect of S1P on *I*_K(Ca)_ in PC12 cells. (**A**) Superimposed current traces obtained in the control and presence of S1P (1 µM). The cells were bathed in a normal Tyrode solution with 1.8 mM CaCl_2_, and the pipette solution containing 0.1 µM Ca^2+^. (**B**) Averaged *I-V* relations of current amplitudes measured at the end of voltage pulses in the absence (◯) and presence (⬤) of 1 µM S1P. Each point represents the mean ± SEM (*n* = 6, in three cultures).

**Figure 7 ijms-22-02175-f007:**
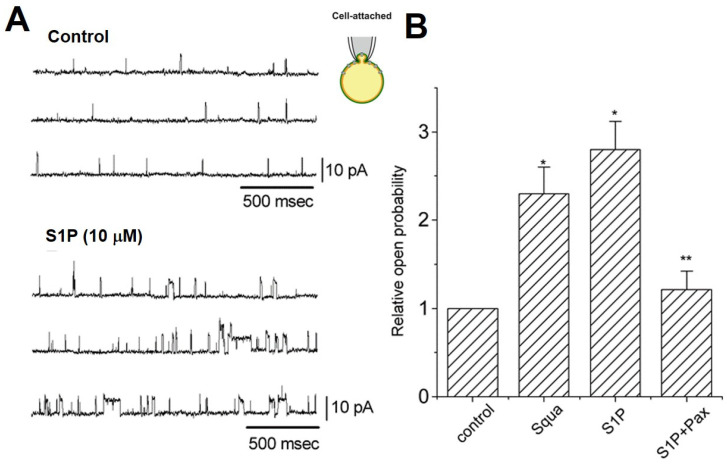
S1P-induced BK_Ca_ activation in the cell-attached configuration of PC12 cells. (**A**) The cells were bathed in a high-K^+^ bathing solution with 1.8 mM CaCl_2_, holding at +60 mV. The original current traces were obtained in the control and 3 min after the S1P (10 µM) application. (**B**) Summary of the effects of squamocin (Squa, 10 µM), S1P, and S1P combined with paxilline (Pax, 1 µM) on the BK_Ca_ open probability. The BK_Ca_ activity in the control was considered to be 1.0 and the relative open probability by each agent was plotted. Asterisks indicate significant differences (*p* < 0.05, * control, and ** S1P alone group, *n* = 5−9, in three cultures).

**Table 1 ijms-22-02175-t001:** The composition of extracellular and intracellular solutions used in this study.

Solution’s Name	Milieu	Composition (in mM)
Normal Tyrode solution	Extracellular	NaCl 136.5, KCl 5.4, CaCl_2_ 1.8, MgCl_2_ 0.53, glucose 5.5, and HEPES 5.5, adjusted with NaOH to pH 7.4
High K^+^-bathing solution	Extracellular	KCl 145, CaCl_2_ 1.8, MgCl_2_ 0.53, and HEPES 5, adjusted with KOH to pH 7.4
K^+^-aspartate solution	Intracellular	K^+^-aspartate 130, KCl 20, KH_2_PO_4_ 1, MgCl_2_ 1, EGTA 0.1, ATP 3, GTP 0.1, and HEPES 5, adjusted with KOH to pH 7.2
High K^+^-pipette solution	Intracellular	KCl 145, MgCl_2_ 2, EGTA 0.1, and HEPES 5, adjusted with KOH to pH 7.2

**Table 2 ijms-22-02175-t002:** Default parameter values used for the modeling of adrenal chromaffin cells.

Symbol	Description	Value
Cm	Membrane capacitance	14 pF
g_Na_	Na^+^ current conductance	65 nS
g_Ca,L_	L-type Ca^2+^ current conductance	0.98 nS
g_Ca,T_	T-type Ca^2+^ current conductance	0.94 nS
g_K(DR)_	K^+^ current conductance	97 nS
g_K(Ca)_	Ca^2+^-activated K^+^ current conductance	0.64 nS
g_K(M)_	M-type K^+^ current conductance	0.43 nS
g_K(erg)_	*erg* K^+^ current conductance	0.95 nS
V_Na_	Na^+^ reversal potential	+60 mV
V_Ca_	Ca^2+^ reversal potential	+100 mV
V_K_	K^+^ reversal potential	−80 mV

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
