# Peer review of "Permissive Modulation of Sphingosine-1-Phosphate-Enhanced Intracellular Calcium on BKCa Channel of Chromaffin Cells"

_ijms, 2021, doi:10.3390/ijms22042175_

Round 1

Reviewer 1 Report

In this study, A. Z. Wu et al. have found that sphingolipid sphingosine-1-phosphate (S1P) modulates the large-conductance Ca2+ and voltage-dependent K+ current (BKCa) in bovine chromaffin cells (BCCs). Such modulation seems to be biphasic: at low concentrations S1P reduces BKCa but at high concentrations (10 μM) BKCa is augmented. This BKCa enhanced stimulation is due to the elevation of the cytosolic Ca2+ concentration monitored with Fura-2, Authors conclude. At the inhibitory concentration (1 μM), S1P increased action potential firing (AP) as expected from enhanced depolarization secondary to BKCa inhibition “By tailoring BKCa channel function, S1P can modulate stimulus-secretion coupling in chromaffin cells, Authors suggest. This is an elegant, well done, and clearly written study. Authors may wish to consider some hints:

  1. The low S1P concentration (1 μM) decreased BKCa and hence, augmented AP firing, as expected from a blocker of outward K+ The high S1P concentration (10 μM) exhibited opposite effects namely, facilitation of BKCa; this should result in a decrease of AP firing rate. Why was this not tested?
  2. Sphingosine can be metabolized to S1P by sphingosine kinases; furthermore, sphingosine was shown to have 4-fold higher potency in triggering endocytosis in BCCs compared with S1P in BCCs (Rosa et al, Pflügers Arch, 2010; 460:901-914). Thus, it could be interesting to check whether sphingosine is more effective in regulating BKCa in BCCs.
  3. The elevation of [Ca2+]c by 10 μM S1P is likely due to ER Ca2+ release; this was shown to be the case in two classical reports that authors may wish to cite (TK Ghosh et al, 1990, Science 248:1653; TK Ghosh et al, 1994, J Biol Chem 269:226-28). A simple experiment is the depletion of the ER Ca2+ with thapsigargin. Under these conditions, the [Ca2+]c elevation elicited by S1P should disappear.
  4. Concerning the regulation of exocytosis by sphingosine and its metabolite S1P, some considerations may be worth of comment:
    • Increase of neurotransmitter release including BCCs has been shown in several laboratories, not cited in the present study (i.e. F. Darios, et al, 2009, Neuron 62:683; JM Rosa et al, 2010, Pflügers Arch 460:901).
    • It could be interesting to test whether S1P augments exocytosis at 1 μM (that increases AP firing) and at 10 μM (that augments [Ca2+]c). This could be easily explored in Author’s lab with capacitance technique.
  5. In the Discussion it is argued that the plasma circulating levels of S1P (0.84 – 4 μM) will saturate its receptors (kDa ≈8-60 nM) (P.9, line 217-226). However, the experiments on isolated outside-out patches suggest a direct action of S1P on BKCa Please explain the relationship (if any) between S1P receptors and the effects of S1P on BKCa at low and high concentrations. The use of S1P receptor blockers could help to clarify this issue.
  6. Minor points:
    • In figure legends it is stated that data points are means ± SEM of n=6-8, n=10 and so on. Please specify whether patches or cell numbers are from 1, 2 or more different cell cultures. This is relevant from a statistical point of view, given the variability of cell parameters among cells from different cultures.
    • P13, l. 403. Please change “Fundin” by “Funding”

Author Response

Thank you for the reviewer's comments, our responses are as the attached file. Please refer.

Reviewer 2 Report

I have enjoyed the opportunity to learn about your results describing the effects of S1P on large conductance calcium activated potassium channels as judged by electrophysiological recordings from chromaffin cells and PC12 cells, complemented in some cases by recordings of changes in intracellular calcium and an attempt to mathematically model S1P effects on induced action potential firing.  Although individual sets of results are novel and somewhat interesting, in combination, this manuscript raises more questions than it answers.  Attention to the following points would significantly improve this paper.

1.  S1P pharmacology - The two interesting patterns of results produced by S1P in this paper are not presented in the context of the extent to which any of the well known S1P receptors are involved.  The authors acknowledge this.  The impact of this paper would be improved significantly with the inclusion of new data obtained in the presence of one or more blockers of S1P receptor subtypes, together with a survey of the PC12 in chromaffin cell literature regarding S1P receptor expression.

2.  Methods - In my experience, preparing S1P in such a way that reliable dose response information can be presented is difficult and requires the use of polar solvents.  Details of S1P solubization must be given.  Second, for each data set the calcium levels or pCa must be given.  In the absence of this, the fact that the macroscopic currents are significantly activated only positive to 0 mV would suggest that there is no physiological significance to the low dose effects reported in your paper.  

3.  The biphasic higher dose effects of S1P are of definite interest but with the exception of the BAPTA data no information is provided concerning how the effects could arise or why they would show what appears to be desensitization.  Further information must be provided.

4.  The single Figure of mathematical modeling that is provided is of interest but it is not sufficient to add any new insights into the phenomenology or attempts to gain mechanistic insight in this paper.  A primary reason is that the induced firing is generated in a cell model that, so far as one knows, has many differences to the chromaffin cell.

5.  The manuscript is presented logically and the Figures have been carefully planned.  In contrast, significant sections of the manuscript, largely due to the use of unconventional Scientific English are ambiguous.  

Examples include: 

Line 40 - 'firings of action potential'

Line 44 - 'may permissively result'

Lines 149 and 150 - these two sentences are unclear

Line 181 - this statement is likely not accurate

Lines 291-293 - your paper presents little fitting information that supports or establishes this contention

Line 384 - Firings of AP impulses?

Line 389 - 'consistently presented'?

Line 403 - 'Fundin'?

Author Response

Thank you for the comments, enclosed please find our response note to each point for your comments. Your favorable consideration is greatly appreciated.

Reviewer 3 Report

The authors describe the effect of sphingosine-1-phosphate on the activity of the large conductance Ca2+-activated K+ channel (BKCa) in catecholamine-secreting chromaffin cells. The manuscript resulted from a detailed and well-conceived study with a clearly defined hypothesis testing mechanisms of action, a clear premise based on published data in other models, and adequate controls, both scientific and technical. The inclusion of pharmacological analyses assessing cellular function as well as meaningful statistical analyses and carefully drawn conclusions add to the high quality of the study.

Author Response

Great thanks for the comments, your favorable consideration is greatly appreciated.

Reviewer: The authors describe the effect of sphingosine-1-phosphate on the activity of the large conductance Ca2+-activated K+ channel (BKCa) in catecholamine-secreting chromaffin cells. The manuscript resulted from a detailed and well-conceived study with a clearly defined hypothesis testing mechanisms of action, a clear premise based on published data in other models, and adequate controls, both scientific and technical. The inclusion of pharmacological analyses assessing cellular function as well as meaningful statistical analyses and carefully drawn conclusions add to the high quality of the study.

Response: The authors thank the reviewers' positive comments and try the best to keep progressive in future studies. Your kind inspiration is greatly appreciated.